# Can Gestural Filler Reduce User-Perceived Latency in Conversation with Digital Humans?

**Junyeong Kum** [1] and **Myungho Lee** [2,*]

1 Department of Information Convergence Engineering, Pusan National University, Busan 46241, Korea
2 School of Computer Science and Engineering, Pusan National University, Busan 46241, Korea
* Correspondence: myungho.lee@pnu.edu; Tel.: +82-51-510-2347

**Abstract:** The demand for a conversational system with digital humans has increased with the development of artificial intelligence. Latency can occur in such conversational systems because of natural language processing and network issues, which can deteriorate the user's performance and the availability of the systems. There have been attempts to mitigate user-perceived latency by using conversational fillers in human–agent interaction and human–robot interaction. However, non-verbal cues, such as gestures, have received less attention in such attempts, despite their essential roles in communication. Therefore, we designed gestural fillers for the digital humans. This study examined the effects of whether the conversation type and gesture filler matched or not. We also compared the effects of the gestural fillers with conversational fillers. The results showed that the gestural fillers mitigate user-perceived latency and affect the willingness, impression, competence, and discomfort in conversations with digital humans.

**Keywords:** virtual reality; virtual human; digital human; conversational filler; gestural filler; latency; embodied conversational agent

## 1. Introduction

The term digital human is generally used to refer to a computer-generated human-like entity. It is often interchangeably used with virtual human (VH) or, with the popularity of the terms metaverse and artificial intelligence (AI), is also called meta-human or AI beings [1]. While the latter two terms arose recently, VH has been used for decades in the virtual reality (VR) community and still is an active research topic [2–5]. In VR, like those other virtual objects, the VH is comprised of computer graphics; however, VHs incorporated with physical body parts (e.g., with mannequins or robots) have also been proposed in the past couple of decades [6–8]. Nevertheless, 3D computer graphics had remained an essential characteristic of VHs when it came to their appearance. On the other hand, in the AI community, instead of 3D modeling and rendering, research on synthesizing human-like entities directly on 2D image or video has been produced [9]. Deep-neural-network-based models not only generate photo-realistic human images, but also synthesize voice, lip sync motion, and gestures [10–12]. Thus, the notion of the VH, which implicates 3D computer graphics, became insufficient to cover those AI-based synthesized humans. To cover both AI-based and 3D-modeled synthesized humans, we use the term digital human throughout this article.

When facing digital humans, users might expect the same social interaction as they would with real humans (RHs), which has been widely investigated with the concept of social presence [13,14]. Making social interaction directly relates to the intelligence of digital humans. Of course, achieving the human level of social interaction is still far from reality. However, in VR, digital humans, i.e., VHs, can be inhabited by users and interact socially with other VHs [15]. Those types of VHs are called avatars, contrary to the agents that are controlled by a computer program. When agents can make conversation with RHs,

they are categorized as conversational agents [16]; furthermore, with the manifestation in the form of a human shape, they are called embodied conversational agents (ECAs) [17]. The digital transformation we are undergoing in the era of Industry 4.0 has enabled the accumulation of big data, and with big data, deep-neural-network-based AI models are widely adopted and improved, making agents rapidly intelligent. In the meantime, such digital transformation also have affected and improved the systems and processes of our society and industry, now demanding the paradigm shift from efficiency- and productivity-centric to human-centric in the upcoming Industry 5.0 and Society 5.0. In Society 5.0, digital humans are expected to co-exist with RHs and be socially more engaging in the form of cobots and/or human digital twins [18,19].

With the recent development of AI models and natural language processing (NLP), the demand for a conversational system with a digital human has increased. There have been attempts to utilize digital humans in various fields, such as health care, psychological consultation, and education [20–22]. Furthermore, researchers have studied ECAs to mimic conversation with an RH [23,24]. These systems often consist of voice processing models, e.g., text-to-speech (TTS) and speech-to-text (STT), and dialogue-generation models. For example, users' utterances are converted into text through an STT model and then analyzed with NLP. Agents then generate responses in text with simple keywords and a sentence-matching-based answer generation [25,26] or a pre-trained language model [27,28], which then is converted into synthesized voice through a TTS model. Some conversational systems with digital humans also include models for generating natural behavior, such as lip sync and gestures [17,24,29–31]. Usually, conversational systems utilize the server–client model. On the client side, a digital human appears on a screen or in a virtual environment, verbalizing the generated sentences and exhibiting gestures. The computationally intensive procedures, e.g., NLP and speech/gesture generations, are processed on the server side and then sent back to the client. Through these processes, the inevitable latency occurs between the user's question and the answer of the digital human [32]. In addition, language models such as GPT3 [33] have limitations in that they can generate ethically inappropriate responses in the given circumstances or answers irrelevant to the user's questions. Therefore, if necessary, a real person has to intervene to provide an appropriate answer to the user's question. Of course, an inevitable time delay occurs in that process of intervention.

The time delay during the conversation with a digital human can reduce the usability of the system and users' satisfaction [34]. While we are unaware of psychophysical experiments investigating the detailed latency requirements in conversation with digital humans, the two-second rule was proposed in man–computer conversational transactions [35], as well as in human–robot interaction; the satisfaction and naturalness decreased after 1 s [36], which is in line with a similar study reporting 0.9 s [37]. Such a short latency allowance seems complicated to achieve considering the current limitations on inference times of TTS, gesture generation, and conversation generation models, even in the case where a computationally powerful server processes all the models and streams the rendered digital humans through a 5G network [38].

Previous studies [39–41] attempted to mitigate user-perceived latency using various conversational fillers, such as "Uhm" or "Wait a minute". However, when it comes to everyday face-to-face communication, people rely not only on verbal cues, but also on nonverbal cues, such as gestures and facial expressions [42–44]; gestures, in particular, have been overlooked. This study, thus, examined whether the gestural filler for digital humans can mitigate user-perceived latency in conversation with a digital human. We compared the effect of conversational fillers and gestural fillers on the user's perception of two conversation types: informative and casual conversation. Considering that speech–gesture match can affect the user's perception [45,46], we also explored the effect of whether the conversation types and gesture fillers matched or not.

In short, this paper addresses the following research questions:

RQ1: Can gestures of digital humans, i.e., gestural fillers, reduce perceived latency in conversation?

RQ2: Does the gesture to reduce perceived latency vary by conversation type?

RQ3: How do gestural and conversational fillers affect users' perception of digital humans when delays occur during the conversation?

The rest of the paper is organized as follows: in Section 2, we summarize previous research on ECAs, the effects of latency in conversation with ECAs, as well as fillers—techniques devised to reduce perceived latency—and the importance of non-verbal cues; in Section 3, we describe the experimental setup and the conversational digital human system we used for the following two studies; in Section 4, we detail the first experiment performed to investigate the correlation between gestural fillers and conversation types; then, in Section 5, we report the second experiment exploring the effects of gestural fillers compared to the conversational fillers; finally, Section 6 concludes the paper and discusses future research directions.

## 2. Related Work

To the best of our knowledge, we are unaware of prior works examining the gestures of digital humans as a means to reduce perceived latency in conversation. While a study investigating a gesture accompanied by a conversational filler has been reported [40], it is unclear if gestures alone could have similar effects. In this section, we present some areas of research that are relevant in various ways.

### 2.1. Embodied Conversational Agents

Text-based chatbots have been developed for various purposes. From simply responding to customers of an online shop to consulting on a financial product to invest in, companies have introduced and applied text-based chatbots for their services. While those chatbots are mainly designed to give information based on users' requests, some are more engaging with users. For example, Woebot [20] detects and diagnoses the depression of users by have intimate conversations with them. Harlie [21] uses the user's speech to diagnose Parkinson's disease. SERMO [22] communicates with users about their emotions and events of daily life through text messages. These chatbot systems are cost-effective and have no constraints on time and space. However, text-based communication is not enough to give the users the same feeling one receives during an actual consultation. Therefore, research on ECAs has increased to compensate for this deficiency.

ECAs have been used as human surrogates and can communicate with users verbally and non-verbally. REA is an embodied virtual real estate agent that responds to users with speech and gestures [24]. In a user study with REA, participants felt more engagement and availability with a responsive agent than with an unresponsive agent. In teaching children about emotions, researchers compared a chatbot using text and an ECA using facial expressions and gestures [29]. The participants chose ECA over the chatbot because it displayed more natural facial expressions, was intelligent and had a personality. Furthermore, learning efficiency was evaluated positively with the ECA. In the SIMSENSEI system [23], users can interact with a virtual human face-to-face for psychological counseling. Furthermore, researchers found that veterans answered more honestly and talked more about their post-traumatic stress disorder (PTSD) symptoms in conditions communicating with virtual humans than with actual counselors [47]. Moreover, having an appearance imposes additional benefits of using ECAs, that is to say, recognizing each as an individual. In a classroom management training system, trainees were exposed to a classroom with multiple VH students [48]. Each of the VHs had a different appearance and voice, thus giving the trainees the feeling of interacting with multiple people, when in fact, a single person controlled all of them [15].

### 2.2. Reducing Perceived Latency in Conversation

Latency in conversation refers to the gap between the question and the answer. *Delay* is often used interchangeably with latency. Research has been conducted on the effect of latency on user perception in human–agent and human–robot interactions. Yang et al. [49]

reported that the frustration and anger of the user increased when a time delay occurred during a task-solving situation with robots. Similarly, the time delay reduced the users' satisfaction and willingness to use the system in the future [34]. Furthermore, researchers compared the effect of time delay (0, 1, 2, and 3 s) in human–robot interaction [36] and found that users considered a delay of 1 s more natural than no delay.

Attempts to mitigate the negative effects of time delay in conversation with digital humans have been continually conducted. The researchers in [39] studied the conversational fillers (CFs) of a robot controlled with the Wizard of Oz method and compared non-CF and CF conditions in open-ended conversation. The users evaluated robots with CF conditions as more alive, human-like, and likable than robots with non-CF conditions. Furthermore, lexical CFs such as "Let me think" were rated higher than non-lexical CFs such as "Hmm." or "Aha.". The researchers in [41] compared uncontextualized and contextualized fillers. With uncontextualized fillers, the digital human says, "Hold on a minute," regardless of the context. With contextualized fillers, the digital human recognizes and reflects on the user's questions to determine what he or she says. For example, if the user asked about the amount of an ingredient, the digital human would say, "Let's see how much you need". In the results, contextualized fillers could mitigate the user-perceived latency more effectively than uncontextualized fillers. Researchers also examined a filler that accompanied both verbal and gestures [50]. In the study, participants watched videos of two digital humans having conversations, in which momentary silents were included. In one condition, digital humans said "Ummm" and touched their own chin during the silence, while exhibiting nothing in another condition. The participants evaluated the silences as less embarrassing when the filler was used.

### 2.3. Non-Verbal Cues in the Conversation

Non-verbal cues, such as eye gaze, facial expressions, gestures, and postures, significantly influence real communication [42]. Not only verbal cues, but also non-verbal cues, including leaning over, nodding gestures, and smiling expressions, could effectively build rapport during communication [43]. Furthermore, gestures of understanding could build a sense of a bond and rapport between clinicians and patients [44].

In some cases, facial expressions and gestures are more effective at delivering information compared to verbal cues. For example, in a study performed by Rogers [51], participants watched videos in which speakers described objects or concepts and were asked to rate their comprehension of the videos. He compared three video representation conditions: audiovisual with lip and facial cues, audiovisual without lip and facial cues, and audio alone. The result demonstrated that visual cues could significantly improve the participants' comprehension and that, with more noise, there was more dependency on visual cues.

Likewise, non-verbal cues play important roles in human–agent and human–robot interaction. Researchers have found that the embodied conversational agent's nodding and glancing influenced the avatar's lifelikeness and fluidity of interaction [30]. Furthermore, users felt more emotional connection and reliability with a virtual human who mimicked the facial expression and intonation of the users [31].

In human–human interaction, speech and the corresponding behaviors occur simultaneously. However, the mismatch between speech and gestures of the speaker can affect the listener's perception. Cassell et al. [45] performed a user study comparing speech–gesture mismatch and match conditions and found that participants considered the inaccuracy of the conversation higher in the mismatch condition. The omission of information was also higher in the mismatch condition. They also reported that listeners identified the fundamental meanings of the utterances through the gestures of the speaker [46].

### 3. Materials

This section describes the physical setup and conversational digital human system we used in the experiments.

We prepared an office-like room consisting of a 65-inch TV, a table, and partitions. A similar virtual environment was implemented and rendered on the TV screen, to mimic a natural face-to-face conversation between two people sitting on each side of the table (see Figures 1 and 2). We used the Unity game engine of version 2019.4.21.f1 (https://unity.com) to render the virtual environment on the TV screen.

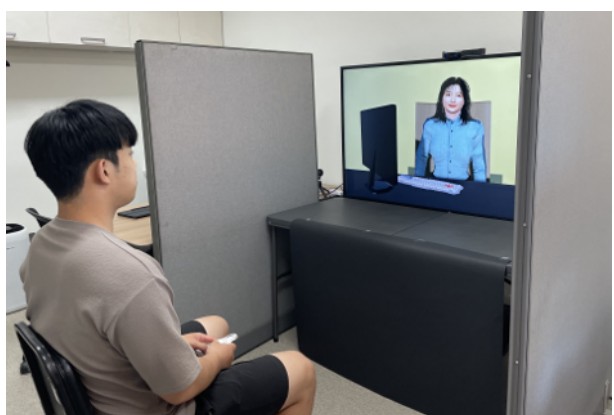

**Figure 1.** The experimental setup for Study 1.

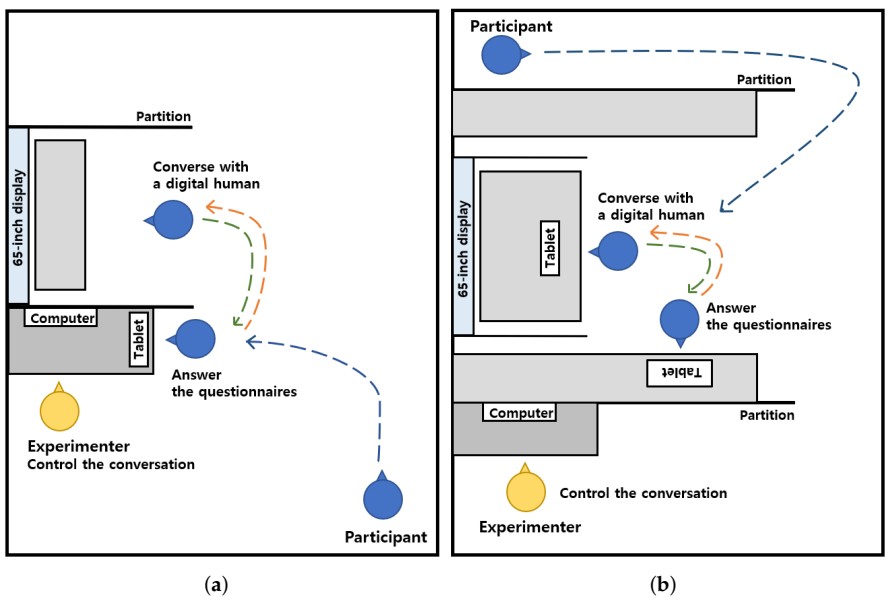

**Figure 2.** The schematics of the experimental space for (**a**) Study 1 and (**b**) Study 2.

Regarding digital humans, we used rigged 3D human models. We used Character Creator 3 (https://www.reallusion.com/character-creator/) to generate the 3D human models. We first collected photos from lab members and used them as the input reference images, from which the software generated the 3D human models. From the generated 3D human models, we modified the appearance of the 3D human models to have different clothing, hairstyles, skin colors, and facial features, but with similar sizes. A total of eight 3D human models were created, four females and four males (see Figure 3). The 3D human models were imported and placed behind a table. with a monitor and keyboard. The monitor and keyboard were placed for one of the gestural fillers used in the experiments.

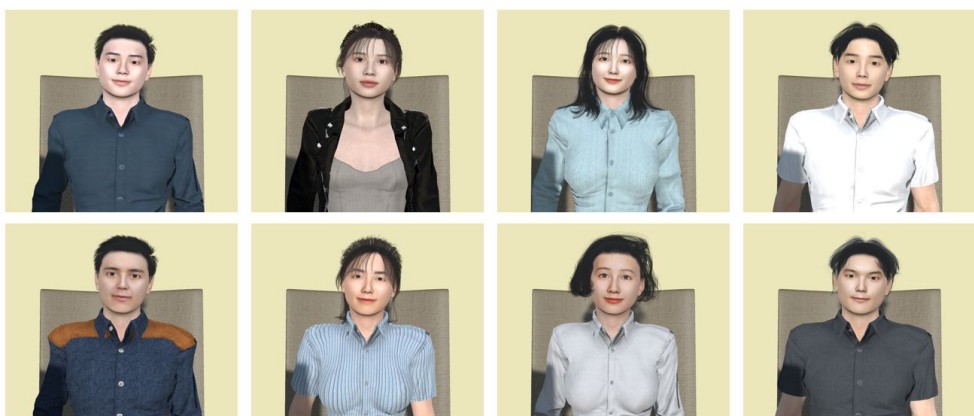

**Figure 3.** The 3D human models used in the studies.

For the conversational ability of the digital human, we exploited the Wizard of Oz paradigm; in other words, an experimenter behind the scene controlled the digital human while users were oblivious to the agency. For that, we designed structured conversations with predetermined question and answer sets, and audio files for the answers were pregenerated in various voices using Typecast (https://typecast.ai/ko). A separate graphical-user-interface (GUI)-based control program was implemented for the experimenter to trigger answers or to change the configuration of the conversational system, including the appearance of the digital human, gestural fillers to exhibit, and voice. Both the control program and the Unity-based digital human rendering program were run on the same local machine to avoid network delay. For natural gaze behavior and lip sync, we used Final IK (http://root-motion.com/) and SALSA LipSync (https://crazyminnowstudio.com/unity-3d/lip-sync-salsa/) In addition, we prepared two gesture animations: a thinking gesture and a typing-a-keyboard gesture (see Figure 4).

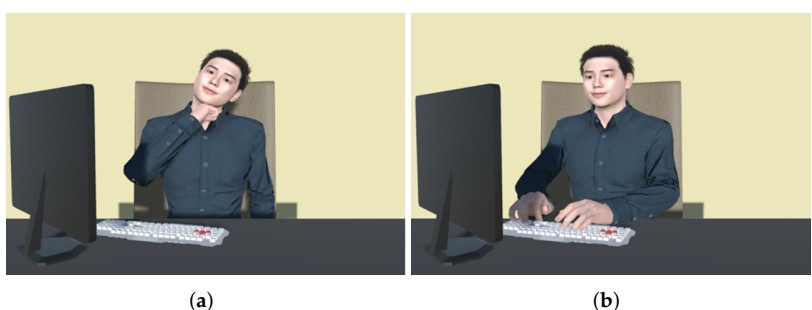

(**a**)　　　　　　　　　　(**b**)

**Figure 4.** (**a**) The thinking motion and (**b**) typing motion.

## 4. Study 1

In this section, we report our user study performed to investigate the effects of congruency or incongruency between conversation types (informative and casual) and gestural fillers (typing and thinking) on the user's perception.

### 4.1. Method

We used a with-in subjects design with four conditions: two mismatch conditions (informative conversation with thinking motion (Figure 4a) and casual conversation with typing motion (Figure 4b)) and two match conditions (informative conversation with typing motion and casual conversation with thinking motion). Two informative conversation sets and two casual conversation sets were used. Each conversation set consisted of eight question–answer turns and did not overlap. Table 1 shows one of the casual and informative conversation sets. Considering the previous study where participants tended to be annoyed regardless of the filler when delays occurred every time [41], we set only four delays out of

eight turns. In every even turns, a latency of 8 seconds occurred, and the digital human behaved according to the conditions. In the conditions with typing motion, the digital human turned his or her body toward the computer and exhibited typing on the keyboard for 8 seconds. After 8 seconds, the digital human turned his or her body back to the user and answered the question. In the conditions with thinking motion, the digital human exhibited thinking motion while touching his or her chin for 8 seconds. In every odd turn without the latency, the digital human answered the question after 0.9 seconds for naturalness [37].

The participants experienced all four conditions, and the order of conditions was counterbalanced and randomized. In all conditions, the participants communicated with four different digital humans (see the first row of Figure 3), and the digital humans' order was also randomized with the Latin square method.

**Table 1.** Conversation examples and the locations of delayed answers.

| | Conversation Type | | Delay | |
|---|---|---|---|---|
| | **Casual** | **Informative** | **Study 1** | **Study 2** |
| 1 | What do you do for fun? | How many cafeterias are there in this campus? | non-delay | non-delay |
| 2 | Do you have any instrument you can play? | How much is breakfast? | delay | |
| 3 | Is there any exercise you do regularly? | How much is lunch in cafeteria? | non-delay | |
| 4 | How do you handle your stress? | Can you tell me the hours of operation of the central library? | delay | |
| 5 | Where was your last overseas trip? | How many books can I borrow at a time in the central library? | non-delay | three delays were randomized |
| 6 | Where do you want to travel abroad the most? | Where is the central library? | delay | |
| 7 | Where is your favorite domestic travel destination? | On which floor can I use a laptop in the library? | non-delay | |
| 8 | Do you have any vacation plans for next summer? | What application should I use to take a seat in the library? | delay | non-delay |

### 4.1.1. Procedure

After a brief explanation, the participant answered a pre-questionnaire using a tablet. The participant sat 1.3 m away from a 65-inch display with a conversation card consisting of eight questions based on the experimental condition (Figures 1 and 2a). For a natural start of the conversation, the digital human stared at a computer first, then turned his or her body after the participant sat down and said, "Hello, nice to meet you. Ask whatever you want".

The participants asked each of the eight questions in order, and the digital human answered according to the conditions. We used the pre-recorded audio files to eliminate unintended latency, which might be caused by NLP or network issues. The experimenter played the appropriate audio file by pressing a button immediately after the participant verbalized the question. After each condition, the participants filled out a post-questionnaire. At the end of the experiment, participants filled out open-ended questions, which included comments for the study, their preferred conditions, as well as the reasons. The whole process was recorded, and informed consent was obtained from all participants involved in the experiment.

### 4.1.2. Measurements

The pre-questionnaire included questions about demographics, prior experiences with digital humans, and the Negative Attitudes towards Robots Scale (NARS) [52]. Prior experiences with digital humans were measured on a 7-point Likert scale (1: not at all, 7:

everyday), and for the NARS, we chose six items relevant to our studies and measured on a 5-point Likert scale, from 1: strongly disagree to 5: strongly agree.

In the post-questionnaire, we measured the following constructs on a 5-point Likert scale (1: strongly disagree to 5: strongly agree) at the end of each condition. The first four were from the research of Boukaram et al. [41], and the latter two were from the research of Carpinella et al. [53]:

- User-perceived latency: We measured how appropriate the response time was for participants in each condition. "The response time of the digital human I just talked to was appropriate".
- Behavioral naturalness: The participants measured the perceived behavioral naturalness of the digital human during the latency. "The gesture of the digital human I just talked to was natural".
- Willingness: The participants evaluated how willing they would be to interact with the digital humans. "I am willing to talk to the digital human I just talked to next time".
- Impression: The participants evaluated the impression of the digital human. "I had a good impression of the digital human I just talked to".
- Discomfort and competence: We used RoSAS [53] to measure these constructs. Each included three questions, and we averaged the ratings per each construct. The discomfort questions asked about the awkwardness, scariness, and strangeness of the digital human. Questions for competence were about the reliability, competence, and interactiveness of the digital human.

### 4.2. Participants

We recruited 14 participants (10 males and 4 females) from a local university who speak Korean as their native language. The average age of the participants was 25.1 (SD = 4.43). The participants had little experience in conversations with digital humans (M = 1.39, SD = 0.86). The participants' majors were diverse, including computer engineering, French, and Philosophy.

### 4.3. Hypotheses

Based on the literature review, we formulated the following hypotheses:

H1   In the casual conversation type, the participants will evaluate the digital human with thinking motion more positively compared to one with typing motion.
H2   In the informative conversation type, the participants will evaluate the digital human with typing motion more positively compared to one with thinking motion.

### 4.4. Results

For discomfort and competence, we used the averaged scores of the ratings for three questions, respectively (Cronbach's $\alpha$ = 0.753 and 0.829). Considering our study design and the ordinal scales of the measures, we performed Friedman tests for each construct and Wilcoxon signed-rank tests for the pairwise comparisons with Bonferroni adjustment, both with a significance level of 0.05. Our main findings are summarized in Figure 5 and Table 2.

We found statistically significant main effects of the conditions on user-perceived latency, behavioral naturalness, and competence. Pairwise comparisons revealed that participants felt thinking gestures were more natural than typing gestures for casual conversation ($p = 0.012$) and that informative conversation with typing gestures was more natural than casual conversation with typing gestures ($p = 0.006$). Regarding digital humans' competence, participants gave higher scores for the digital humans who exhibited typing gestures for informative conversation compared to those who exhibited typing gestures during casual conversation ($p = 0.002$). While a significant main effect of the conditions on user-perceived latency was reported, the post hoc test did not find any differences between conditions.

We further analyzed the data by grouping them over the match and mismatch conditions. Casual thinking and informative typing were considered as match conditions

and the other two as mismatch conditions. Wilcoxon signed-rank tests were performed on the measures at a 5% significant level. The results showed that there were statistically significant differences between match and mismatch conditions on user-perceived latency ($p = 0.004$) and behavioral naturalness ($p = 0.004$).

Regarding the NARS, we averaged the ratings (Cronbach's $\alpha = 0.671$) and performed Pearson correlation tests between the NARS score and each construct; however, we did not find any significant correlations.

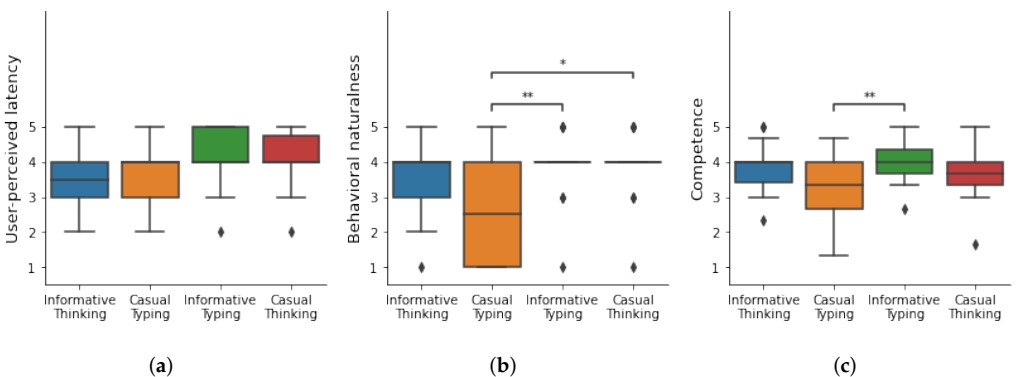

**Figure 5.** The results of (**a**) user-perceived latency, (**b**) behavioral naturalness, and (**c**) competence. (*: $p < 0.05$, **: $p < 0.01$)

**Table 2.** Summary of the Friedman test results.

|  | $\chi^2$ | $p$-Value |
|---|---|---|
| User-perceived latency | 9.246 | 0.026 |
| Behavioral naturalness | 13.720 | 0.003 |
| Willingness | 1.316 | 0.725 |
| Impression | 0.721 | 0.868 |
| Competence | 13.925 | 0.003 |
| Discomfort | 1.250 | 0.741 |

*4.5. Discussion*

Overall, our results indicate whether conversation and gesture types are matched or not has a strong effect on the perception of latency in conversation with digital humans. Participants were more tolerant of the delays towards digital humans who exhibited appropriate gestures. However, the appropriateness of the gesture varied by the conversation.

In this study, we divided question–answer types of conversation into two groups: informative and casual. In the casual category, questions were related to one's own information, thoughts, or past experiences; in contrast, questions in the informative conversation type were related to objective facts, such as bus fare or a library's closing time. In other words, if the answerer were an actual human, answers for casual conversation could be obtained by thinking, and for informative conversation, answers could be acquired by searching external sources, such as the Internet. Interestingly, participants, in general, expected similar behavior from digital humans (Figure 5b, Table 2). In casual conversation, participants felt digital humans exhibiting the thinking gesture were more natural than ones with typing, i.e., searching the Internet gesture (partially supporting H1). However, the differences between the gestures in perceived naturalness were not significant in informative conversation. This might be related to how participants considered digital humans, i.e., whether they thought of digital humans as social beings or mere technology. Participant P1's comment, *"I thought that the digital human has all of the information, so he pretends to*

*think"*, is in line with this speculation. Similar to P1, some participants might think that the digital human knows everything, thus retrieving the information internally might seem better. Therefore, for them, the digital being using the Internet, i.e., external sources, to find information might seem awkward.

Regarding user-perceived latency, the mismatch between conversation and gesture might have triggered the recognition of the delays and/or reassessment of digital humans (Figure 5a, Table 2). As sensorimotor incongruency leads to breaks in presence in immersive virtual environments, inappropriate gestures could have led to breaks in social presence with digital humans, therefore reducing their tolerance of delays. It could have made participants focus—otherwise distracted—on the delay moments. While we are oblivious to the reasons yet, our results strongly indicate that participants felt less delay when digital humans exhibited appropriate gestures, i.e., matched with the context.

In the competence of the digital human, regardless of the gestural filler, the informative conversation type had a higher score than the casual conversation type (Figure 5c, Table 2). The reason seems to be that the informative conversation type is based on objective facts, contrary to the casual conversation type.

## 5. Study 2

In this section, we present our second study performed to compare the effects of conversational and gestural fillers on users' perception of the digital human and the latency in conversation. We again used two conversation types: informative and casual, and chose the appropriate gestural filler per each conversation type based on the results of Study 1. The filler types compared in this study were non-filler, conversational filler only, gestural filler only, and gestural filler accompanied by conversational filler (NF, CF, GF, and GCF, respectively).

### 5.1. Method

We used a with-in subjects design with two independent variables: conversation type and filler type. Each participant experienced a total of eight conditions (two conversation types × four filler types). We used the typing and thinking motions as gestural fillers for informative and casual conversation, respectively. Considering that the lexical conversational filler was better than the non-lexical conversational filler at mitigating user-perceived latency [39], we used "I am searching for information" and "Please wait a minute" as the conversational fillers for informative and casual conversation, respectively. The duration of the utterances for each sentence was approximately the same (2 s) in Korean. The conversational fillers were initiated three seconds after the participants' utterances ended (Figure 6).

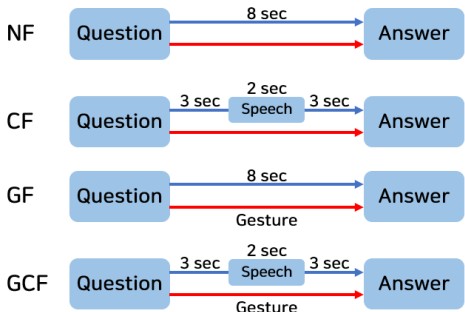

**Figure 6.** Experimental conditions (NF, CF, GF, and GCF).

Four informative conversation sets and four casual conversation sets were used. Each conversation set consisted of eight question–answer turns and did not overlap. To prevent participants from recognizing patterns of latency occurrence, the order of delay and non-delay turns was randomized. We set the first and last turns to be non-delay turns, and in

between, we randomly assigned three delay turns, but not three consecutive times. In the non-delay turns, the digital human answered the question after 0.9 s for naturalness [37]. In the delay turns, the digital human exhibited gestures according to the condition for eight seconds. The participants experienced all eight conditions, and in each condition, they interacted with different digital humans (Figure 3). The orders of conditions and digital humans were counter-balanced and randomized with the Latin square method.

### 5.1.1. Procedure

In Study 1, we observed that some participants kept looking downward, where they held the conversation card, instead of looking at the digital human and that some lost track of the order of questions. To address these issues, we placed a table on the participants' side and let them hold a tablet PC above the table. The tablet PC displayed one question at a time (Figures 2b and 7).

The rest of the procedure was the same as in Study 1 (cf. Section 4.1.1).

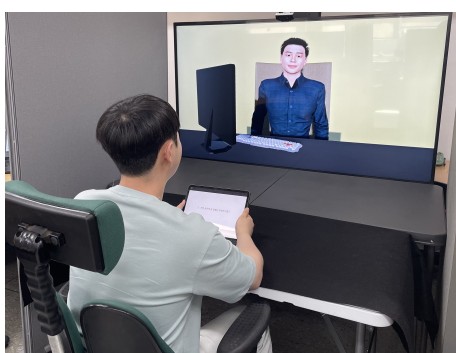

**Figure 7.** The experimental setup.

### 5.1.2. Measurements

We used the same pre- and post-questionnaires used in Study 1 (cf. Section 4.1.2). In addition, we analyzed how much the participants looked at the digital human during the eight seconds of latency from the recorded videos.

### 5.2. Participants

A total of 32 participants (10 males and 22 females) recruited from a local university volunteered in this study (mean age: 21.88, SD = 2.56). All of them speak Korean as their mother tongue, and they had little or no prior experience in conversation with digital humans (M = 1.17, SD = 0.57). We excluded those who participated in Study 1.

### 5.3. Hypotheses

Based on the literature review and our study design, we formulated the following hypotheses:

H1 The CF, GF, and GCF conditions will receive higher scores than NF for user-perceived latency.

H2 The GF and GCF conditions will receive higher scores than CF and NF for behavioral naturalness.

H3 The CF and GCF conditions will receive higher scores than GF and NF for willingness and impression.

H4 The GCF and GF conditions will receive higher scores than CF and NF for discomfort.

### 5.4. Results

For discomfort and competence, we used the averaged scores of the ratings for three questions, respectively (Cronbach's $\alpha$ = 0.733 and 0.780). To compare the effects of fillers, we used mean ratings over conversation types per filler condition per participant in this analysis. We performed Friedman tests on each subjective measure and used Wilcoxon

signed-rank tests for pairwise comparisons with Bonferroni adjustment applied to the p-values. The significance level was set to 0.05 for all statistical analyses. Our results are summarized in Table 3 and Figure 8.

We found statistically significant main effects of filler types on all measures. We present the summarized results of pairwise comparisons in connection with the hypotheses in the following itemized list:

- User-perceived latency : There were significant differences between NF and other fillers (CF, GF, and GCF > NF, in all $p < 0.001$), supporting H1. We also found a significant difference between CF and GCF ($p = 0.036$). However, the results did not reveal any statistical differences between GF and GCF and between GF and CF (See Figure 8a and Table 3).
- Behavioral naturalness : Pairwise comparisons revealed that there were statistically significant differences between NF and GF ($p < 0.001$), NF and GCF ($p < 0.001$), CF and GF ($p < 0.001$), and CF and GCF ($p < 0.001$), supporting H2. No statistical differences were found between NF and CF and GF and GCF (See Figure 8b and Table 3).
- Willingness : We found significant differences between NF and other conditions ($p = 0.025$, $p < 0.001$, $p < 0.001$, respectively, in the order of CF, GF, GCF), between CF and GF ($p = 0.004$), and between CF and GCF ($p = 0.021$). There was no statistical difference between GF and GCF. These results contradict our hypothesis H3 (See Figure 8c and Table 3).
- Impression : Our post hoc tests indicated statistically significant differences between NF and GF ($p < 0.001$), NF and GCF ($p < 0.001$), CF and GF ($p < 0.001$), and CF and GCF ($p = 0.011$). We, however, did not find differences between NF and CF and GF and GCF. The results for impression also do not support H3 (See Figure 8d and Table 3).
- Discomfort : Statistically significant differences were found between NF and GF ($p < 0.001$), NF and GCF ($p = 0.031$), and CF and GF ($p = 0.001$). In general, NF and CF received higher scores compared to GF and GCF, partially supporting H4. We did not find any statistically significant differences between NF and CF, CF and GCF, and GF and GCF (See Figure 8e and Table 3).
- Competence : Although we did not set any hypotheses on competence with regard to filler conditions, our results revealed statistically significant differences between NF and GF ($p < 0.001$), NF and GCF ($p < 0.001$), CF and GF ($p = 0.004$), and CF and GCF ($p = 0.022$). In general, participants considered digital humans competent for the conditions accompanying gestural fillers (See Figure 8f and Table 3).

With regard to the participants' gaze behavior during the delayed responses, one of the experimenters went over the recordings frame by frame and calculated the average time looking at the digital human per each filler condition. Although the nature of the time is an interval, our data failed to pass the Shapiro–Wilk normality test; thus, we decided to perform a non-parametric Friedman test for the gaze behavior. The Friedman test revealed a statistically significant main effect of filler types on gaze behavior ($\chi^2 = 28.809$, $p < 0.001$). The Wilcoxon signed-rank test with Bonferroni correction performed on each pair showed statistically significant differences between NF and GF ($p < 0.001$), NF and GCF ($p < 0.001$), CF and GF ($p = 0.004$), and CF and GCF ($p = 0.017$). However, statistically significant differences between NF and CF and GF and GCF were not found (see Figure 9).

No statistically significant correlations were found between the NARS and measures; however, there was a slight tendency for discomfort (Pearson's $r = 0.338$, $p = 0.058$). Participants with higher NARS scores tended to rate higher discomfort for digital humans.

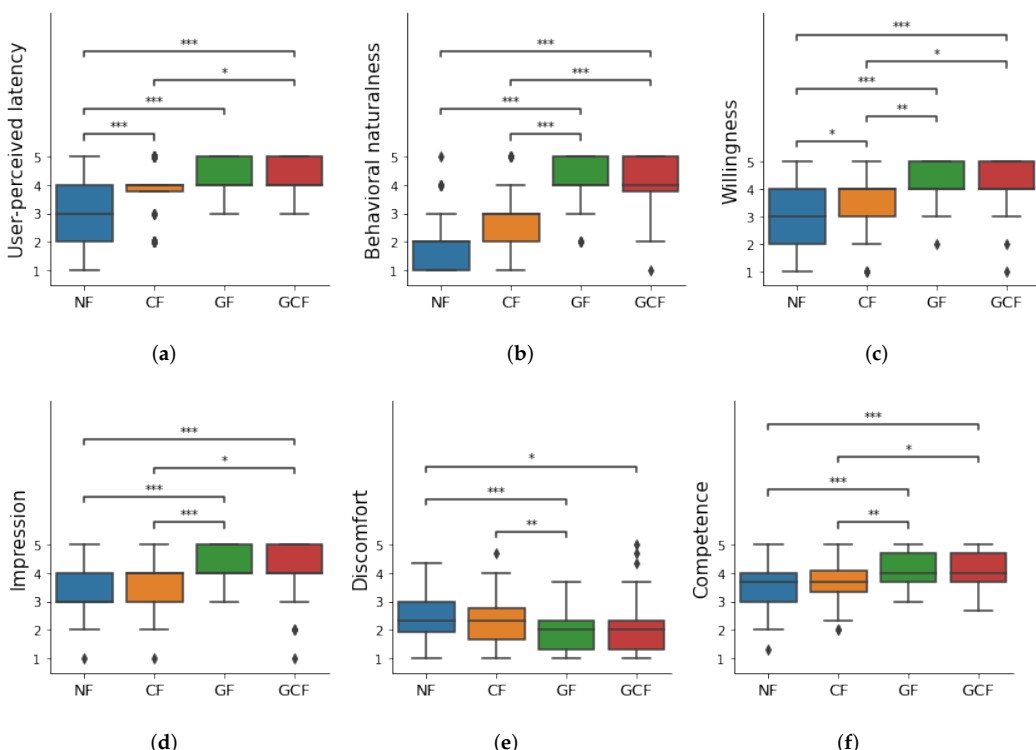

**Figure 8.** Results of the subjective measures: (**a**) user-perceived latency, (**b**) behavioral naturalness, (**c**) willingness, (**d**) impression, (**e**) discomfort, and (**f**) competence. (*: $p < 0.05$, **: $p < 0.01$, ***: $p < 0.001$).

**Table 3.** Summary of the Friedman test results.

|  | $\chi^2$ | *p*-Value |
|---|---|---|
| User-perceived latency | 59.477 | <0.001 |
| Behavioral naturalness | 66.031 | <0.001 |
| Willingness | 50.109 | <0.001 |
| Impression | 42.301 | <0.001 |
| Discomfort | 22.510 | <0.001 |
| Competence | 40.208 | <0.001 |

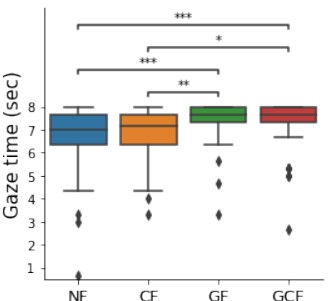

**Figure 9.** Results of the gaze time during the delayed responses. (*: $p < 0.05$, **: $p < 0.01$, ***: $p < 0.001$).

*5.5. Discussion*

Our results show that CF, GF, and GCF can mitigate user-perceived latency more than NF (supporting H1). In the previous study, there were attempts to reduce the perception of latency using conversational fillers of digital humans [39,41]. We found that the gestural

filler can also effectively mitigate user-perceived latency (Figure 8a, Table 3). Even though the latency was eight seconds in all conditions, the participants felt the length of latency differently. For the NF condition, participant P17 commented *"The response time was so slow that I didn't feel like a real conversation."*. Regarding GF condition, participant P22 mentioned *"I think the answer time was faster than other digital humans."*. In the delay situations, the digital human acting dormant, could lead to breaks in social presence of the participants, therefore increasing their perceived latency.

According to behavioral naturalness, our results statistically support H2 (Figure 8b, Table 3). For CF condition, P13 said, *"When the digital human says, "Please wait a minute." I was a little distant. It would be more natural with a gesture or facial expression"*. Similar to P13, some participants for NF commented that they felt awkward and that it would be better for digital humans to take some action than just staring ahead. Furthermore, some participants for the CF and GCF conditions commented that it would be more natural if the digital human says things like *"Hmm.."* instead of *"Please wait a minute"* or *"I am looking for information"*. These comments differ from previous studies' results that the lexical conversational filler is better than the non-lexical conversation filler [41]. It seems that more research related to the conversation filler of digital humans is needed.

For willingness, the GF and GCF conditions received statistically higher scores than NF and CF (Figure 8c, Table 3). The participants evaluated that they would like to talk more with digital humans using the gestural filler in the future. Participant P29 for GCF commented *"I felt like I was having a real conversation. I want to talk to her more"*. It seems that gesture fillers can improve the usability of digital humans.

Regarding impression, the GF and GCF conditions received significantly higher evaluation than NF and CF (Figure 8d, Table 3). The participants had a good impression of the digital humans with gestural fillers. Participant P20 for the NF condition said *"Unlike other digital humans, it was a little scary to say nothing and blink when the delay occurred"*. However, P20 and P31 for the GCF condition commented *"I felt like he was likable and friendly."* and *"I felt like I was talking about a vacation plan with my real friend."*, respectively. This is in line with the previous study that non-verbal cues play an important role in building rapport and bonding during the conversation [43,44]. Furthermore, it is one of the main advantages of embodied conversational agents that they can utilize their bodies like actual humans.

Our results partially support H4. The participants felt more discomfort with digital humans in the NF than the GF and GCF conditions (Figure 8e, Table 3). In the NF condition, the participants could have experienced breaks in social presence due to digital humans who did not take any action during the latency, unlike real people. Participant P15 commented *"I felt uncomfortable when he stared at me doing nothing"*.

The participants evaluated the digital humans using the gestural filler as more competent (Figure 8f, Table 3). In the informative conversation type, it seems that the participants considered the digital humans smart enough to use a computer. Participant P4 said *"I think she did not understand me because she did not take any action saying, "I am looking for information." I felt a little repulse."* for the CF condition. The participants were confused about whether the digital human understood their question when only conversational fillers were provided.

The participants focused visually on the digital human for GF and GCF more than NF and CF (Figure 9). It can be assumed that the gestural filler, in which the digital human takes specific actions, drew the participant's visual focus, rather than the conversational filler or non-filler, in which the digital human does nothing outwardly. However, there was no significant difference between the questionnaires and the gaze time of each participant.

## 6. Conclusions

In this paper, we demonstrated the potential of gestural fillers when latency occurs in the conversation with embodied conversational agents, i.e., digital humans. In Study 1, we examined whether the congruency between the gestural fillers and conversation types affects the user's perception of the embodied conversational agent. In two congruency

conditions (informative conversation with typing motion and casual conversation with thinking motion), the participants evaluated the digital humans as more natural, as well as felt less latency, addressing RQ2. Furthermore, in Study 2, we further compared the effects of conversational and gestural fillers on the user's perception and gaze behavior, which gave answers to our RQ1 and RQ3. Our results indicated that gestural fillers could mitigate user-perceived latency more effectively than conversational fillers and make digital humans' behavior more natural. In addition, the gestural fillers positively affected users' willingness to talk to and impression of the digital humans, as well as their perceived competence, while reducing the discomfort the users felt. It goes without saying that gestural fillers were more effective in drawing users' attention than conversational fillers.

While several interesting findings were drawn, there also are a few limitations to be noted. First, the digital human repeatedly exhibited the same gestural and conversational fillers per condition. For example, we used thinking motion with "Wait a minute" in the casual conversation type and typing motion with "I am searching for information" in the informative conversation type. Some participants pointed out that, *"The digital human acted with the same pattern in three out of eight turn-takings."* and she/he also stated *"It was a little boring."* and *"If the digital human made various motions or used different words, the conversation would be less boring."*. To address this issue, we plan to build a gesture-generation model to generate appropriate gestural fillers based on the given conversation and context. Furthermore, the participants pointed out that the digital humans' intonation, gestures, and facial expressions were unnatural compared to real people.

Those unnatural intonations, gestures, and facial cues might have been more substantial than latency; thus, further comparison studies should be conducted for better usability of conversational digital humans. Regarding latency, we utilized pre-determined scripts and pre-recorded audio files to exclude unintended latency factors, e.g., networking, inference time, etc. Those omitted factors should be mathematically modeled to determine when to initiate fillers in practice. Finally, all participants were Koreans; thus, the cultural diversity of the participants was limited. The gestures and utterances used in this study can be perceived differently outside Korea, similar to the meaning of nodding one's head in India differs from most other countries. Therefore, future work should consider including participants from various cultural backgrounds.

**Author Contributions:** Conceptualization, J.K. and M.L.; Data curation, J.K.; Formal analysis, J.K. and M.L.; Funding acquisition, M.L.; Investigation, J.K.; Methodology, J.K. and M.L.; Software, J.K.; Visualization, J.K.; Writing—original draft, J.K.; Writing—review & editing, M.L. All authors have read and agreed to the published version of the manuscript.

**Funding:** This research is supported by Year 2021 Culture Technology R&D Program by Ministry of Culture, Sports and Tourism and Korea Creative Content Agency (Project Number: R2021040269).

**Institutional Review Board Statement:** The study was conducted in accordance with the Declaration of Helsinki, and approved by the Institutional Review Board of Pusan National University (protocol code 2022-76-HR, 3 August 2022).

**Informed Consent Statement:** Informed consent was obtained from all subjects involved in the study.

**Data Availability Statement:** Not applicable.

**Conflicts of Interest:** The authors declare no conflict of interest.

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
