# Peer review of "Can Gestural Filler Reduce User-Perceived Latency in Conversation with Digital Humans?"

_applsci, doi:10.3390/app122110972_

Round 1

Reviewer 1 Report

In general, the quality of research done is good while the quality of the presentation needs to be improved.

In the state of the art it could be useful to include more references that analyse the introduction of with conversational agents from the perspective of end users like for example: Rampioni M., Stara V., Felici E., Rossi L., Paolini S. Embodied conversational agents for patients with dementia: Thematic literature analysis (2021) JMIR mHealth and uHealth, 9 (7),  DOI: 10.2196/25381

It could be useful to explain in a clearer way at the beginning the structure of the paper. For example: the first part is generic on conventional agents and use of different types of fillers. So, the reader wonder why in the first study wonders why only gestural filers are analysed.

Likewise, in the introduction different types of conversational agents and robot are mentioned but in the studies are used specific systems whose description is in my opinion not sufficient (details only on the tools used to create them)

In general, the condition of experimental setup and protocol are not clearly described. It is necessary to explain better what you have done.

For example:

·         The description of the procedure of the experiment is distributed in different point in the text and it is quite difficult to understand it.

·         It is not clearly explained if the eight questions were fixed or selected in a list or free, It would be useful to include a list of the question asked during experimental phase.

·         It not really clear if the latency-no latency approach was applied always to the same questions or was random

·         The conversational agent was the same in study one and two and for all the participant to each study?

There are some typing errors:

·         In Row 105 probably there is a type because the authors list three different alternative but the second and third alternatives are the same.

·         At row 165 there is an incongruency: 5-point Likert scale (1: strongly disagree to 7: strongly agree)

·         The two hypothesis (rows 181-185) contains some text errors

Some comments related to methodology and discussion:

·         The number of participants has been justified by a methodological evaluation of sample size, especially for study 1 that has a very low sample size?

·         Why only in study two was investigated about their negative attitudes toward robots

·         Have you evaluated the bias related to young age and high level of education of participants?

·         The graphs in figure 3 are not really informative. A line graph is typically used for (and suggest) a trend analysis while in this case you are comparing just two alternatives.

Author Response

The major changes are outlined below. To help you parse this response, we use abbreviations for comments (e.g. C1 : comment 1). We also use the symbols ### throughout to mark the start of our response. 

C1 : “It could be useful to include more references that analyse the introduction of with conversational agents from the perspective of end users.”

### In response to the reviewer’s advice, we revised the introduction to include descriptions of virtual human and embodied conversational agents. Also, we added recent research trends of virtual humans.

C2 : “Explain in a clearer way at the beginning the structure of the paper.”

### We added the structure of the paper at the end of Section 1 Introduction.

C3-1 : “The description of the procedure of the experiment is distributed in different point in the text and it is quite difficult to understand it.”

### As the reviewer pointed out, the description of the experimental procedures were scattered in Section 4 and 5. We improved the paper by: (1) modifying section 4.1.1. and 5.1.1. Procedure, (2) adding Figure2.

C3-2 : “It is not clearly explained if the eight questions were fixed or selected in a list or free, It would be useful to include a list of the question asked during experimental phase.”

### We added Table 1, which shows examples of informative and casual conversation sets we used in the experiments.

C3-3 : “It’s not really clear if the latency-no latency approach was applied always to the same questions or was random.”

### In Study 1, we assigned delay turns in every even turn and non-delay turns in every odd turn. In Study 2, we fixed the first and last questions as non-delay turns, and assigned delay turns randomly. We supplemented this explanation in Section 4.1. and 5.1. To facilitate understanding, we added an example of conversation sets and delay assignments for Study 1 and Study 2 separately in Table 1.

C3-4 : “The conversational agent was the same in study one and two and for all the participant to each study?”

### In Study 2, we used additional four digital humans in addition to four digital humans used in Study 1. We supplemented this explanation in Section 4.1. and 5.1.   

C4-1 : In Row 105 probably there is a type because the authors list three different alternative but the second and third alternatives are the same.

### We modified this error.

C4-2 :  At row 165 there is an incongruency: 5-point Likert scale (1: strongly disagree to 7: strongly agree)

### We modified this error.

C4-3 : The two hypothesis (rows 181-185) contains some text errors

### We fixed this error.

C5 : The number of participants has been justified by a methodological evaluation of sample size, especially for study 1 that has a very low sample size?

### Considering the small sample size, we re-analyzed the data with a non-parametric method.

C6 : Why only in study two was investigated about their negative attitudes toward robots

 ### It was initially omitted in Study 1 to reduce the number of pages. We revised the paper to included initially omitted results as well as clarified measures for both studies.

C7 : Have you evaluated the bias related to young age and high level of education of participants?

### We evaluated the bias related age and gender. There was no statistically significant impact of those factors in Study 1 and Study 2. The participants had similar educational backgrounds (undergraduate students and graduate students), so we did not compare them separately.

C8 : The graphs in figure 3 are not really informative. A line graph is typically used for (and suggest) a trend analysis while in this case you are comparing just two alternatives.

### We changed figure 3 to boxplots (Figure 5). Also, we added a statistical analysis in the result table (Figure5, 8, 9)

Reviewer 2 Report

This is a well-written and interesting study that addresses an important but often neglected aspect of ECAs. My specific comments follow.

page 3 - verbal&bodily fillers is a strange concept. Do you mean verbal and non-verbal fillers/behaviours? Please describe. Did that work find that verbal or bodily was better or that combined was best?

'facial expressions in case." this does not make sense - is the sentence finished? Do you mean in some cases?

page 5 - 5-point Likert scale (1: strongly disagree to 7: strongly agree). - this is a 7 point likert scale. You have stated this twice please check and correct.

Do you address the gender imbalance in your participants. 14 participants is very small.

How were participants recruited? Who are these participants? What are they studying? Are they postgraduates? Was there ethics approval?

H2 In the informative conversation type, In the casual conversation type, the participants - this seems to be an error. I assume you forgot to delete the second clause.

page 6 - "We found that whether the conversation type and gestural filler congruence can affect the user-perceived latency.' This sentence needs correcting.

page 11 - I wonder if there are some culture specific findings in this paper. If the study was done with students from other countries would they consider "hmm" to be more acceptable than saying "please wait a minute". If it was a voice/text only conversational agent then I would imagine "please wait a minute" would be more meaningful.

page 12 - sample size and only use of Korean students could also be seen as a limitation for the external validity of the results.

Author Response

The major changes are outlined below. To help you parse this response, we use abbreviations for comments (e.g. C1 : comment 1). We also use the symbols ### throughout to mark the start of our response. 

C1 : “verbal&bodily fillers is a strange concept. Do you mean verbal and non-verbal fillers/behaviours? Please describe. Did that work find that verbal or bodily was better or that combined was best?”

### We revised the paragraph with more detailed explanation of the condition they used in their experiment.   

C2 : “'facial expressions in case." this does not make sense - is the sentence finished? Do you mean in some cases?”

### We modified this sentence for the correct expression.

C3 : “5-point Likert scale (1: strongly disagree to 7: strongly agree). - this is a 7 point likert scale. You have stated this twice please check and correct.”

### We modified this error.

C4 : “Do you address the gender imbalance in your participants? 14 participants is very small.”

How were participants recruited? Who are these participants? What are they studying? Are they postgraduates? Was there ethics approval?

### First of all, the study was approved by IRB at our institution, we added a sentence at the end of the procedure section to clarify it as recommended by MDPI. We evaluated the bias related to age and gender, but we couldn’t find any statistically significant impacts of the two factors in Study 1 and Study 2. The participants had similar educational backgrounds (undergraduate and graduate students), so we did not compare them separately. We recruited participants via the university website.

C5 : “H2 In the informative conversation type, In the casual conversation type, the participants - this seems to be an error. I assume you forgot to delete the second clause.”

### We fixed this error.

C6 : "We found that whether the conversation type and gestural filler congruence can affect the user-perceived latency.' This sentence needs correcting.”

### We fixed the ambiguity of this sentence.

C7 : “I wonder if there are some culture specific findings in this paper. If the study was done with students from other countries would they consider "hmm" to be more acceptable than saying "please wait a minute". If it was a voice/text only conversational agent then I would imagine "please wait a minute" would be more meaningful.”

C8 : “sample size and only use of Korean students could also be seen as a limitation for the external validity of the results.”

### For the above two, we agree with your points. Our participants lack diversity in cultural backgrounds, thus we mentioned it as a limitation of our study at the end of the paper. Regarding the sample size, we reanalyzed the data with non-parametric tests.